# Altered Left Ventricular Rat Gene Expression Induced by the Myosin Activator Omecamtiv Mecarbil

**DOI:** 10.3390/genes14010122

**Published:** 2023-01-01

**Authors:** Bachar El Oumeiri, Laurence Dewachter, Philippe Van de Borne, Géraldine Hubesch, Christian Melot, Pascale Jespers, Constantin Stefanidis, Kathleen Mc Entee, Frédéric Vanden Eynden

**Affiliations:** 1Department of Cardiac Surgery, Université Libre de Bruxelles (ULB) Erasme University Hospital, 1070 Brussels, Belgium; 2Laboratory of Physiology and Pharmacology, Faculty of Medicine, Université Libre de Bruxelles (ULB), 1070 Brussels, Belgium; 3Department of Cardiology, Université Libre de Bruxelles (ULB) Erasme University Hospital, 1070 Brussels, Belgium

**Keywords:** left ventricle, omecamtiv mecarbil, gene expression, apoptosis, metabolism, oxidative stress

## Abstract

To explore the impact of omecamtiv mecarbil (OM) on the gene expression profile in adult male rats. Fourteen male Wistar rats were randomly assigned to a single OM (1.2 mg/kg/h; n = 6) or placebo (n = 8) 30-min infusion. Echocardiography was performed before and after OM infusion. Seven days after infusion, rats were euthanized, and left ventricular (LV) tissues were removed for *real-time* quantitative polymerase chain reaction (RTq-PCR) experiments. After OM infusion, pro-apoptotic *Bax*-to-*Bcl2* ratio was decreased, with increased *Bcl2* and similar *Bax* gene expression. The gene expression of molecules regulating oxidative stress, including glutathione disulfide reductase (*Gsr*) and superoxide dismutases (*Sod1*/*Sod2*), remained unchanged, whereas the expression of antioxidant glutathione peroxidase (*Gpx*) increased. While LV gene expression of key energy sensors, peroxisome proliferator activator (*Ppar*) α and γ, AMP-activated protein kinase (*Ampk*), and carnitine palmitoyltransferase 1 (*Cpt1*) remained unchanged after OM infusion, and the expression of pyruvate dehydrogenase kinase 4 (*Pdk4*) increased. The LV expression of the major myocardial glucose transporter *Glut1* decreased, with no changes in *Glut4* expression, whereas the LV expression of oxidized low-density lipoprotein receptor 1 (*Olr1*) and arachidonate 15-lipoxygenase (*Alox15*) increased, with no changes in fatty acid transporter *Cd36*. An increased LV expression of angiotensin II receptors *AT1* and *AT2* was observed, with no changes in angiotensin I-converting enzyme expression. The Kalikrein-bradykinin system was upregulated with increased LV expression of kallikrein-related peptidases *Klk8*, *Klk1c2*, and *Klk1c12* and bradykinin receptors B1 and B2 (*Bdkrb1* and *Bdkrb2*), whereas the LV expression of inducible nitric oxide synthase 2 (*Nos2*) increased. LV expression in major molecular determinants involved in calcium-dependent myocardial contraction remained unchanged, except for an increased LV expression of calcium/calmodulin-dependent protein kinase II delta (*Cacna1c*) in response to OM. A single intravenous infusion of OM, in adult healthy rats, resulted in significant changes in the LV expression of genes regulating apoptosis, oxidative stress, metabolism, and cardiac contractility.

## 1. Introduction

Heart failure (HF) is a major cause of morbidity and mortality and remains a public health problem worldwide [1,2]. Current therapies used to treat HF, including β-blockers, diuretics, and angiotensin-converting enzyme (ACE) inhibitors, are not completely effective. Similarly, while inotropic drugs, including β-adrenergic agonists (e.g., dobutamine) and phosphodiesterase inhibitors (e.g., levosimendan and milrinone) increase cardiac output (CO), their ongoing use results in increased myocardial oxygen consumption, high levels of intracellular calcium (Ca^2+^), elevated heart rate, arrhythmias, and mortality [3]. Indeed, these drugs are known to increase the rate of Ca^2+^ cycling and ATP utilization in the myocardium because more Ca^2+^ needs to be removed from the cytoplasm and sequestered in the sarcoplasmic reticulum [4]. The adverse long-term effects of these drugs may be related to the actions of Ca^2+^/calmodulin-dependent protein kinase II (CaMKII) and other protein kinases that induce myocardial apoptosis, hypertrophy, and fibrosis [5]. Similarly, β-adrenergic activation alters myocardial metabolic substrate use and may trigger energy deficits and oxidative stress [6].

Omecamtiv mecarbil (OM) is a novel small molecule that directly activates cardiac myosin. The sarcomere contains both thin and thick filaments and is the fundamental unit of cardiac muscle contractility. Cardiac myosin, which is the main component of the thick filament, uses chemical energy derived from ATP hydrolysis to generate contractile force. OM selectively activates the S1 domain of cardiac myosin, but has no impact on myosin filaments from any other muscle [7]. The administration of OM increases the rate of ATP turnover and improves contractility by increasing the number of myosin heads capable of interacting with actin filaments, albeit with no impact on Ca^2+^ homeostasis [7]. In a canine model of systolic HF, OM increased the systolic ejection time, stroke volume (SV), and CO [8]. In vitro and in vivo studies confirmed that OM selectively inhibits cardiac myosin ATPase and, thus, has the potential to decrease myocardial oxygen consumption [9]. Contrarily, the administration of OM increased myocardial oxygen consumption in a pig model of HF [10] and resulted in impaired myocardial efficiency by increasing O_2_ consumption both at baseline and during work in an isolated mouse heart model; these responses were abolished by the addition of a myosin-ATPase inhibitor [10]. While these data suggested that the administration of OM may result in increased O_2_ consumption, these findings were not fully consistent with the OM-mediated inhibition of baseline myosin ATPase activity observed in vitro [9]. Similarly, Nagy et al. [11] reported that OM-treated myofilaments were sensitized to Ca^2+^ in an exposed rat myocyte model, while Utter et al. [12] found that OM re-sensitized myofilaments exhibited decreased Ca^2+^ sensitivity in a mouse model of dilated cardiomyopathy. 

In this context, the objective of this study was to examine the OM-mediated changes in gene expression in adult rat myocardium, with a particular emphasis on the pathways associated with apoptosis, oxidative stress, energy substrate metabolism, and Ca^2+^-mediated cardiac contractility. 

## 2. Methods

### 2.1. Animal Model and Experimental Design

The experimental protocol was approved by the Institutional Animal Care and Use Committee of the Faculty of Medicine of the Université Libre de Bruxelles (ULB; Brussels, Belgium; protocol acceptation number: 644N). Experiments were conducted in accordance with the Guide for the Care and Use of Laboratory Animals published by the United States National Institutes of Health (NIH Publication No. 85-23; revised 1996). 

Fourteen adult male Wistar rats (Janvier, Le Genest-Saint-Isle, France) were randomly assigned for intravenous administration of OM (1.2 mg/kg/h for 30 min via the femoral vein; n = 6; mean body weight: 553 ± 38 g) or placebo (n = 8; mean body weight: 536 ± 39 g) on day 0. Dose of OM was chosen to achieve peak plasma concentrations of ~400 ng/mL, as previously reported [13]. Seven days after OM infusion, OM- and placebo-treated rats were sacrificed by exsanguination via section of the abdominal aorta. The hearts were rapidly harvested and dissected to isolate the LV, which was snap-frozen in liquid nitrogen and stored at −80 °C for further biological analysis.

### 2.2. Echocardiography and Cardiac Measurements

Transthoracic 2D, M-mode, and Doppler echocardiography were performed using an ultrasound scanner (Vivid-E90, GE Healthcare, Wauwatosa, WI, USA) equipped with a 12-MHz phased-array transducer (GE 12S-D, GE Healthcare) in anesthetized rats (with inhaled 1.5% isoflurane). All echocardiographic measurements were obtained by the same observer, according to the American Society of Echocardiography guidelines [14]. Standard right parasternal (long and short axis) and left apical parasternal views were used for data acquisition. Fractional shortening (FS) was calculated using the formula (FS = LVEDD − LVESD/LVEDD × 100) in M-mode from a LV short-axis view. Ejection fraction (EF) was derived using the Teicholz formula. Electrocardiogram was monitored via limb leads throughout the procedure. Aortic flow was measured from the left apical view to calculate forward stroke volume (SV) and cardiac output (CO) and to measure pre-ejection period (PEP: delay from Q wave of QRS to aortic opening; ms), LV ejection time (LVET: interval from beginning to termination of aortic flow; ms), and inter-beat interval (RR; ms). Systolic time (ms) was determined as PEP + LVET (ms), and diastolic time (ms) was calculated as RR interval − systolic time. Echocardiography was performed before and 30-min after OM/placebo administration.

### 2.3. Real-Time Quantitative Polymerase Chain Reaction (RTq-PCR)

Total RNA was extracted from snap-frozen LV myocardial tissue using TRIzol reagent (Invitrogen, Merelbeke, Belgium), followed by a chloroform/ethanol extraction and a final purification using QIAGEN RNeasy^®^ Mini kit (QIAGEN, Hilden, Germany), according to manufacturer’s instructions. RNA concentration was determined by standard spectrophotometric techniques, using a spectrophotometer Nanodrop^®^ (ND-1000; Isogen Life Sciences, De Meern, The Netherlands), and RNA integrity was assessed by visual inspection of GelRed (Biotium, Hayward, California)-stained agarose gels. Reverse transcription was performed using random hexamer primers and Superscript II Reverse Transcriptase (Invitrogen, Merelbeke, Belgium), according to the manufacturer’s instructions. Gene-specific *sense* and *antisense* primers for RTq-PCR (Table 1) were designed using the Primer3 program for *rattus norvegicus* gene sequences, including those for B-cell lymphoma 2 (*Bcl2*), Bcl2 associated X apoptosis regulator (*Bax*), glutathione peroxidase (*Gpx*), glutathione-disulfide reductase (*Gsr*), superoxide dismutases 1 and 2 (*Sod1* and *Sod2*), AMP-activated protein kinase (*Ampk*), peroxisome proliferator-activated receptors α and γ (*Ppar α* and *γ*), solute carrier family 2 members 1 (*Slc2a1*, also known as *Glut1*) and 4 (*Slc2a4* or *Glut4*), pyruvate dehydrogenase kinase (*Pdk4*), carnitine palmitoyltransferase1 (*Cpt1*), fatty acid transporter *Cd36*, oxidized low-density lipoprotein receptor 1 (*Olr1*, also known as *Lox1*), arachidonate 15-lipoxygenase (*Alox15*), angiotensin II receptor type 1a (*Agtr1a*, also known as *AT1*), angiotensin II receptor type 2 (*Agtr2*, also known as *AT2*), angiotensin-converting enzymes 1 and 2 (*ACE1* and *ACE2*), nitric oxide synthases 2 and 3 (*Nos2*, also called *inducible NOS* or *iNOS* and *Nos3*, also called *endothelial NOS* or *eNOS*), kallikrein-related peptidases 8 and 10 (*Klk8* and *Klk10*), kallikrein 1-related peptidases C2 and C12 (*Klk1c2* and *Klk1c12*), bradykinin receptors B1 and B2 (*Bdkrb1* and *Bdkrb2*), ATPase sarcoplasmic/endoplasmic reticulum Ca^2+^ transporting 2 (*Atp2a2*, also called *Serca2*), ryanodine receptor 2 (*Ryr2*), calcium voltage-gated channel subunit alpha1C (*Cacna1c*), solute carrier family 8 member A1 (*Slc8a1*), Ca^2+^/calmodulin-dependent protein kinase II delta (*Camk2d*), glyceraldehyde-3-phosphate dehydrogenase (*Gapdh*), and hypoxanthine phosphoribosyltransferase 1 (*Hprt1*) used as housekeeping genes. Intron-spanning primers were selected whenever possible to avoid inappropriate amplification of contaminant genomic DNA. Amplification reactions were performed in duplicate using SYBRGreen PCR Master Mix (Quanta Biosciences, Gaithersburg, MD, USA), specific primers, and diluted template cDNA. Analysis of the results was performed using an iCycler System (BioRad Laboratories, Hercules, CA, USA). Relative quantification was achieved using the Pfaffl method [15] by normalization with the housekeeping genes, *Gapdh* and *Hprt1*.

### 2.4. Statistical Analysis

Results are presented as mean ± standard deviation (SD), with “n” representing the number of individual data points. The echocardiographic data were compared using Student’s *t*-test for repeated measures (n = 6) before and after OM perfusion. The RTq-PCR data evaluating LV gene expression were compared using Student’s t-test for independent samples (with n = 8 in the control group and n = 6 in the treated group). Statistical analyses were performed using StatView 5.0 software. A *p*-value < 0.05 was considered statistically significant.

## 3. Results

### 3.1. Effect of OM on Cardiac Function in Rats

As illustrated in Table 2, the infusion of OM increased the fractional shortening (FS) and the LV ejection time (LVET) and decreased the aortic pre-ejection period (PEP), which resulted in a reduction in the PEP/LVET ratio. No other investigated echocardiographic parameters were significantly altered by OM infusion (Table 2). 

### 3.2. OM Altered Myocardial LV Expression of Genes Regulating Apoptosis and Oxidative Stress

Myocardial LV gene expression of anti-apoptotic *Bcl2* was significantly higher in rats treated with OM, compared to placebo, whereas no difference in gene expression of pro-apoptotic *Bax* was observed (Figure 1A). The resulting pro-apoptotic *Bax*-to-*Bcl2* ratio was reduced in the LV of rats that underwent OM infusion (Figure 1A). We also examined the differential expression of genes involved in oxidative stress regulation. The myocardial expression of *Gpx*, an antioxidant enzyme, increased in the LV after OM infusion, whereas no changes in *Gsr*, *Sod1*, or *Sod2* gene expression were observed (Figure 1B).

### 3.3. OM Impacted LV Expression Profile of Key Determinants of Cardiac Energy Substrate Use

To assess the effects of OM infusion on basal myocardial energy metabolism, we evaluated the gene expression profile of the transcription factors and molecules regulating cardiac glucose and fatty acid metabolism. As illustrated in Figure 2A, myocardial LV gene expression of key energy sensors *Ppar α* and *γ* and *Ampk* remained unchanged. Myocardial LV expression of *Slc2a1* (*Glut1*), the major myocardial glucose transporter, decreased after OM infusion, while gene expression of *Slc2a4* (*Glut4*) remained unchanged (Figure 2B). In contrast, myocardial LV expression of *Pdk4*, a mitochondrial pyruvate dehydrogenase (PDH) regulator overarching metabolic shift between fatty acid oxidation and glycolysis as energy fuel, increased after OM infusion (Figure 2C), whereas the carnitine palmitoyltransferase1 (*Cpt1*) remained unchanged (Figure 2D). As illustrated in Figure 2E, the OM infusion increased the LV expression of *Alox15* encoding the 12/15 lipoxygenase enzyme implicated in polyunsaturated fatty acid metabolism and of oxidized low-density lipoprotein receptor 1 (*Olr1*, also known as *Lox1*) encoding for a scavenger receptor mediating the uptake of oxidized lipoproteins into cells, whereas the gene expression of fatty acid transporter *Cd36* remained unchanged.

### 3.4. OM Altered LV Expression of Genes Implicated in Cardiac Contractility

Because OM is a myosin-specific activator that increases myocardial contractility independently of Ca^2+^ fluxes, we evaluated the OM-induced myocardial expression of different regulators of cardiac contraction. As illustrated in Figure 3A, OM infusion increased the myocardial LV gene expression of both angiotensin receptors *AT1* and *AT2*, while the gene expression of angiotensin-converting enzymes *ACE1* and *ACE2* remained unchanged (Figure 3B). Myocardial LV expression of NO-synthase catalyzing the production of NO, a key modulator of myocardial function, was increased by OM infusion for the inducible *iNOS* isoform, while it remained stable for the constitutive *eNOS* isoform (Figure 3c). The kallikrein-bradykinin system was upregulated in the LV of rats after OM infusion, with the increased myocardial LV expression of genes encoding the serine proteases Klk8, Klk1c2, and Klk1c12 (Figure 3D), as well as the bradykinin receptors (Bdkr) B1 and B2 (Figure 3E), which are G-protein-coupled receptors mediating kinin actions. No change in myocardial gene expression in *Klk10* was observed (Figure 3D). Finally, OM infusion did not induce any changes in the gene expression of major players involved in Ca^2+^-dependent cardiac contraction, except for an increase in LV gene expression of *Cacna1c* in response to OM (Figure 3F).

## 4. Discussion

The present results show that a single 30-min infusion of myosin activator OM induced significant LV expression alterations in genes regulating apoptosis (with decreased pro-apoptotic *Bax*-to-*Bcl2* ratio), oxidative stress (with increased antioxidant *Gpx*), cardiac metabolism (with decreased *Glut1* and increased *Lox1*, *Alox15*, and *Pdk4*), and contraction (with increased *AT1* and *AT2*, upregulation of kallikrein-bradykinin system, but no changes in molecules involved in Ca^2+^-dependent myocardial contraction) 7 weeks after OM infusion. 

In the present study, we evaluated the echocardiographic parameters and gene expression in the LV of rats after OM infusion, compared to placebo-infused rats. As previously reported [8,16], the administration of OM resulted in increases in both the ejection time and the FS. Our findings did not achieve statistical significance for the ejection fraction, potentially because of the limited number of rats and/or the concentration of drug used. However, we did not observe significant increases in CO and SV in OM-treated rats. 

The first set of gene expression profile focuses on the differential expression of the genes regulating apoptosis. Specifically, we examined the OM-induced gene expression of mitochondrial anti-apoptotic Bcl2, and pro-apoptotic Bax [17]. Bcl2 is known to control the release of cytochrome c, preserve mitochondrial integrity, and protect against apoptosis [18]. Our findings revealed a significant increase in *Bcl2* expression, leading to a down-regulation of the *Bax*-to-*Bcl2* ratio, in response to OM infusion. The *Bax*-to-*Bcl2* ratio reflects an overall vulnerability to apoptosis; increases in the *Bax-to-Bcl2* ratio suggest higher levels of apoptotic activity [19]. In contrast to the response to the inotropic agent dobutamine, which partially activates the apoptosis processes in vivo [20], our results suggested that OM did not activate apoptotic processes and was even able to protect the LV against apoptosis. Gpx is one of three main antioxidant enzymes [21]. Our findings revealed that OM infusion resulted in increased expression of *Gpx*, but had no impact on any other tested antioxidant genes. Interestingly, these findings are contrasted with those reported for other inotropic drugs. Indeed, levosimendan was shown to reduce oxidative stress through increased expression of genes encoding *Sod* and *Gpx* [22]. In a recent publication, Rhoden et al. [23] reported that OM promoted the accumulation of mitochondrial reactive oxygen species (ROS) in both rat and human cardiac tissues. As ROS are known to promote the expression of *Gpx* [24], our results suggested a link between OM and this critical antioxidant pathway.

We also evaluated the effects of OM on cardiac metabolism via the evaluation of genes involved in the metabolism of fatty acids, glucose, and lactate and the production of high-energy phosphates [25]. Glut1 is the major determinant of homeostatic glucose transport in cardiac muscle [25]. Administration of OM resulted in decreased expression of *Glut1* in rat LV, which may result in decreased glucose uptake. In contrast to the findings reported for dobutamine [26], the present results suggest that OM promoted a shift from glycolytic to oxidative metabolism [26]. Fatty acid catabolism is coordinately regulated with glucose pathways to support homeostasis. in response to changes in energy supply or demand. Reciprocal regulation in fatty acid and glucose metabolism involves both the PDH complex and the Cpt [27]. PDH converts the pyruvate generated by glycolysis to acetyl-CoA and CO_2_ via oxidative decarboxylation, while the Cpt contributes to fatty acid transport into mitochondria, where they undergo oxidation to generate acetyl-CoA [27]. Pdk4 is an important regulator of PDH activity [28]. Increased levels of Pdk4 promote the inactivation of PDH and, thus, act on the metabolic shift from glucose to fatty acid oxidation [28]. In the present study, the administration of OM resulted in increased expression in *Pdk4*, which may be a marker of increased fatty acid oxidation in the LV [29].

Decreased glucose uptake and increased fatty acid oxidation may result in the production of higher levels of ATP and higher O_2_ consumption. These findings are consistent with those reported by Bakkehaug et al. [10], who reported that the administration of OM resulted in increased myocardial oxygen consumption. Similarly, Lox1, which was originally identified as a receptor for oxidatively-modified LDL [30], can be induced by numerous stimuli, including angiotensin II [31], shear stress [32], and ischemia-reperfusion injury [33]. Alox15, which is a lipid-peroxidizing enzyme [34], has been implicated in the pathogenesis of atherosclerosis [30,31,35], diabetes, and neurodegenerative disease [34]. The expression levels of both *Lox1* and *Alox15* were markedly increased in HF [35,36], and increased expression of *Lox1* was detected in cases of diastolic dysfunction [24,37]. OM-associated diastolic dysfunction and stiffness have also been reported [38,39]. Here, we found increased LV expression in *Lox1* and *Alox15* in OM-infused rats. Because *Lox1* expression has been related to diastolic dysfunction, the link between OM infusion, specific molecular determinants, and diastolic dysfunction should be further studied in future studies.

Angiotensin II binding to AT1 induces vasoconstriction and promotes oxidative stress by activating NADPH oxidase and inducing eNOS uncoupling. This results in a switch from NO to the production of ROS, including superoxide. In contrast, binding to AT2 promotes vasorelaxation, protection against ischemia-reperfusion injury and myocardial infarction, and decreased inflammation [40]. Thus, the activation of AT2-mediated pathways may counter-regulate those resulting from AT1 activation [41]. Here, we found increased gene expression of both *AT1* and *AT2* in response to OM, with an observed *AT2*-to-*AT1* ratio of 1.15. A high *AT2*-to-*AT1* ratio has been associated with increased oxidative stress and cardiac cell apoptosis [42]. The relatively low ratio observed in the present study is consistent with the absence of activation of apoptosis and oxidative stress. Nitric oxide synthase (NOS) catalyzes the conversion of L-arginine to L-citrulline and NO, which is a free radical involved in both homeostatic and immunological functions. iNOS is a Ca^2+^-independent enzyme that is expressed in cardiomyocytes, in response to environmental perturbation (e.g., cytokine release) [43]. The activation of iNOS results in substantially higher levels of NO, compared to other forms of NOS [44]. In the heart, iNOS contributes to a contractile dysfunction characteristic of ischemia-reperfusion injury, infarction, and HF [45,46]. In contrast, several studies have shown beneficial effects of iNOS in the normal, hypertrophied, transplanted, or cardiomyopathic human heart [47,48]. Here, we found that the administration of OM resulted in increased *iNOS* expression, whose significance to OM mechanism of action has to be further determined.

KLK8 has been previously detected in the rat myocardium [49]. Although its physiologic substrates remain largely unknown, the expression of *Klk8* protects against acute ischemia-reperfusion injury and induces cardiac hypertrophy in rats [49,50], in response to pressure overload [49]. Klk1c2 (also known as tonin) can catalyze the release of angiotensin II directly from angiotensinogen; thus, the activation of this enzyme may result in increased production of angiotensin II, independently of ACE activity [51]. Klk1c2 may also induce cardiac hypertrophy [52]. Klk1c2 perfusion in Wistar rats induced coronary vasoconstriction and simultaneously depressed myocardial contractility; the time to peak for cell shortening and half relaxation was significantly reduced. All these results suggest that Ca^2+^ handling is significantly accelerated by Klk1c2 [53]. Direct interactions between Kllk1c12 or Klk1 and the Kinin B2 receptor are critical factors responsible for cardioprotective responses. The activation of this pathway is known to inhibit oxidative stress, apoptosis, and inflammation, as well as cardiac hypertrophy and fibrosis [54]. These ligand-receptor interactions result in improved cardiac function and lead to reduced blood pressure [54]. In our study, OM administration resulted in an increased expression of *Klk1*, *Klk1c2*, and *Klk8*. In the present study, increases in all these three proteases may allow the heart to develop a combined adaptative response to OM [50]. The bradykinin receptor family includes two G protein-coupled receptors (Bdkrb1 and Bkrbr2) that also mediate the biological effects of kinins [55]. Bdkrb1 is expressed and synthesized *de novo*, in response to tissue injury and inflammation [56]. Bdkrb2 is the main receptor for bradykinin; it interacts directly with AT2 [56], as well as other receptors [55]. While signaling, both Bdkrb1 and Bdkrb2 induce NO production [55], and their overall physiological and pathophysiological significance remain unknown [55]. Cardioprotective effects mediated by Bdkrb1 or Bdkrb2 alone via ACE-inhibition have been reported [55]. Endothelial overexpression of *Bdkrb1* in rat models has resulted in an expanded LV cavity and reduced function [57]. Bradykinin-mediated upregulation of Bdkrb2 in the absence of Bdkrb1 did not provide full cardioprotection. Interestingly, the upregulation of Bdkrb1, in the absence of Bdkrb2, results in further tissue damage [58]. In the present study, the administration of OM resulted in increased expression of both *Bdkrb1* and *Bdkrb2* and could, therefore, suggest potential cardioprotective effects. 

The L-type Ca^2+^ channel α1C-subunit gene (*Cacna1c*) plays an essential role in cardiac excitation–contraction coupling [59]. This protein is localized in the T-tubule sarcolemma, adjacent to RYR2, where it controls Ca^2+^ influx from the extracellular milieu into the cytosol and, thus, serves as a major determinant of cardiac function. β-adrenergic receptor stimulation increases the number of L-type channels at the sarcolemma, which results in enhanced Ca^2+^ influx and amplification of excitation–contraction coupling [60]. Prolonged AT1 signaling via reduced L-type Ca^2+^ channels results in a negative inotropic effect [60]. Ca^2+^-calmodulin-dependent protein kinase II (CaMKII) activity controls the expression of *Cacna1c* in isolated rat neonatal ventricular cardiomyocytes [38]. In the present study, the administration of OM increased the expression of *Cacna1c*, although this resulted in no modification of the *Serca2*, *RYR2*, *GLUT1*, or *CamkII* expressions; these have been all implicated in maintaining Ca^2+^ homeostasis. In canine LV monocytes, OM has been recently shown to affect intracellular Ca^2+^ homeostasis by increasing the capacity of RYR2 to remain open, therefore impacting cardiomyocyte repolarization [61]. However, it seems that *Cacna1c* would provide only minor contributions to intracellular Ca^2+^ release. Administration of OM did not result in a significant increase in the concentration of cytosolic Ca^2+^. The increase in *Cacna1c* expression in response to OM might, instead, be related to a diastolic Ca^2+^ leak from the sarcoplasmic reticulum [38]. Low-level Ca^2+^ release induced by Caν1.2 α1 may serve to restore and maintain Ca^2+^ levels that resulted from the leak in the SR [62]. As previously reported [39], permeabilized human cardiomyocytes exhibited a marked reduction in the rate of force generation and relaxation once the Ca^2+^ concentration reached a steady state in permeabilized human cardiomyocytes. Collectively, these findings suggest the existence of a previously unidentified action of OM in promoting Ca^2+^ regulation at the actin-myosin complex. 

The present study has several limitations. Although the effects of OM are concentration-, time-, and species-dependent [23], only one set of experimental conditions was examined here. Furthermore, the experimental data were obtained in healthy rats. Thus, caution is appropriate when extrapolating these data to humans. In addition, the evaluation was performed at gene level and may not correspond directly with protein levels *in vivo*. Future studies should be performed in experimental models with LV pathology to better understand the effects of OM in this context. 

In conclusion, OM infusion in rats resulted in a gene expression profile suggesting myocardial LV preservation against apoptosis and oxidative stress, together with an increased fatty acid oxidation that may be compatible with an increased O_2_ consumption rate. Interestingly, the administration of OM did not induce any changes in the LV expression of the genes involved in Ca^2+^ homeostasis and its associated contraction. It will be critical to design future studies built on these findings to understand the precise mechanisms of action underlying OM.

## Figures and Tables

**Figure 1 genes-14-00122-f001:**
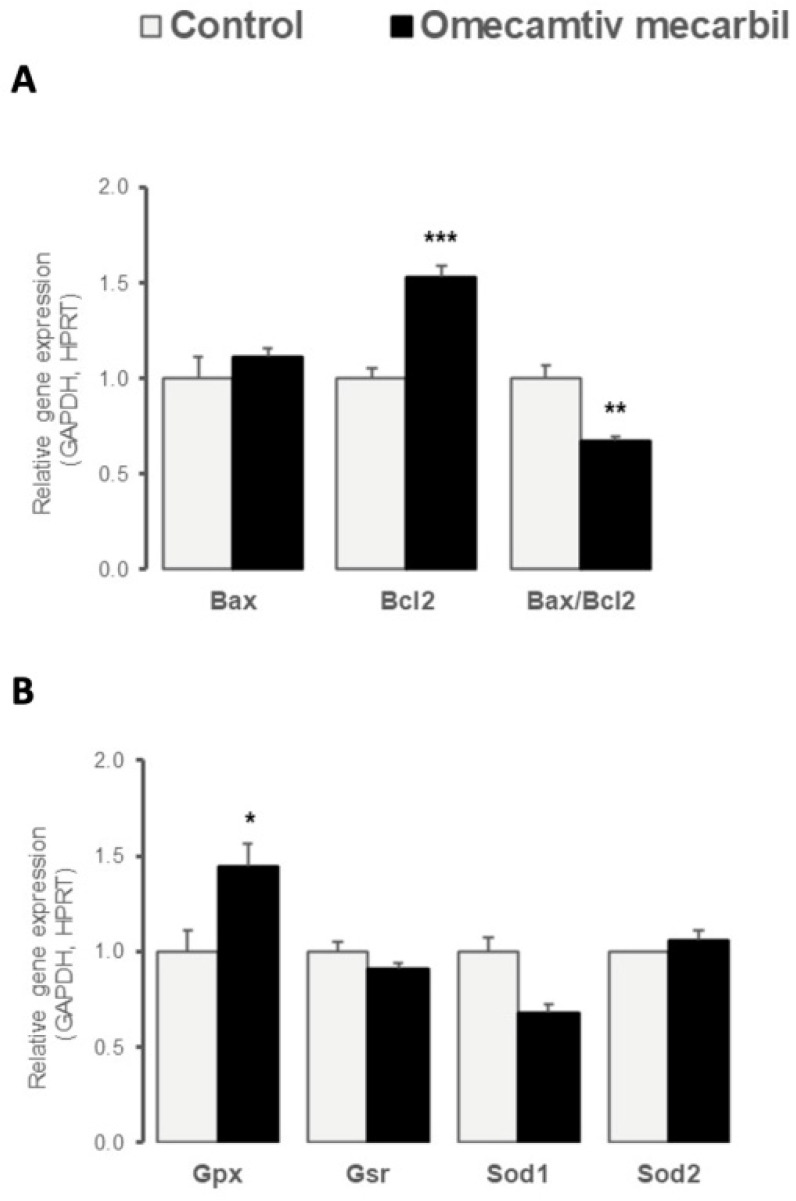
Myocardial left ventricular relative expression of genes implicated in (**A**) apoptosis (*Bax*, *Bcl2*) and (**B**) oxidative stress (*Gpx*, *Gsr*, *Sod1*, *Sod2*) processes seven days after omecamtiv mecarbil (OM; n = 6; black bars) versus placebo (n = 8; grey bars) infusion. Values are presented as mean ± SD; * 0.01 < *p* < 0.05, ** 0.001 < *p* < 0.01, *** *p* < 0.001.

**Figure 2 genes-14-00122-f002:**
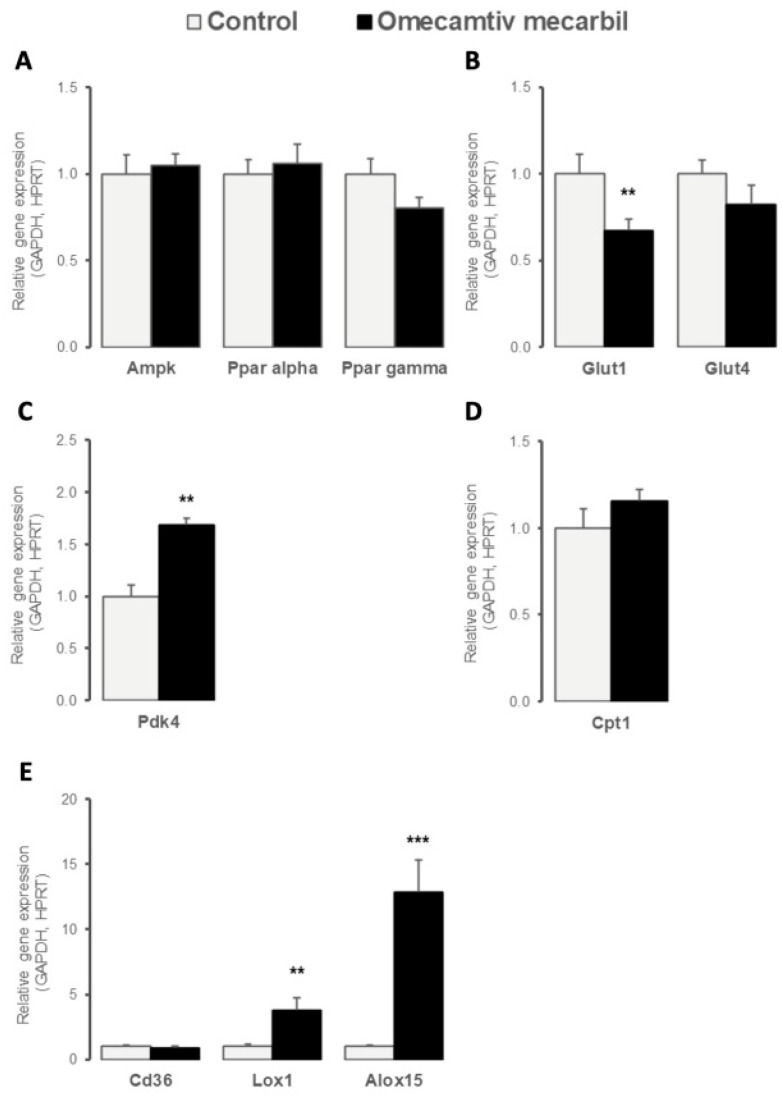
Myocardial left ventricular relative expression of genes implicated in cardiac metabolism, including (**A**) cellular energy sensors such as *Ampk*, *Ppar α*, and *Ppar γ*; (**B**) glucose transporters *Glut1* and *Glut4*; (**C**) mitochondrial metabolic regulators contributing to glucose to fatty acids shift as cardiac major energy fuel, such as *Pdk4* and (**D**) *Cpt1*; and (**E**) fatty acid metabolism regulators such as *Cd36*, *Lox-1*, and *Alox-15*, seven days after omecamtiv mecarbil (OM; n = 6; black bars) versus placebo (n = 8; grey bars) infusion. Values are presented as mean ± SD; ** 0.001 < *p* < 0.01, *** *p* < 0.001.

**Figure 3 genes-14-00122-f003:**
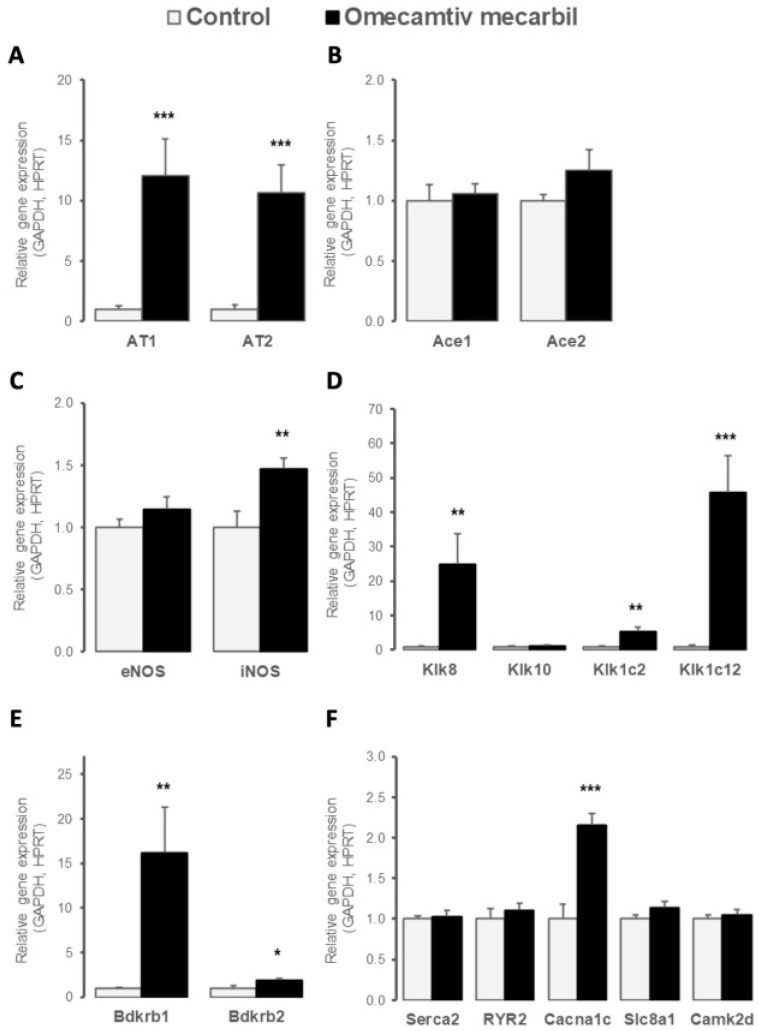
Myocardial left ventricular relative expression of genes controlling myocardial contractility including (**A**) *AT1* and *AT2* angiotensin II receptors; (**B**) *ACE1* and *ACE2* angiotensin-converting enzymes; (**C**) endothelial (*eNos* or *Nos3*) and inducible (*iNos* or *Nos2*) nitric oxide synthases; (**D**) major cardiac actors of kallikrein (*Klk*8, *Klk*10, *Klk*1c2, and *Klk*1c12)-(**E**) bradykinin (*Bdkrb*1 and *Bdkrb*2) system and of (**F**) Ca^2+^-dependent excitation–contraction *Atp2a*, *Ryr2*, *Cacna1c*, *Slc*8a1, and *Camk2d* seven days after omecamtiv mecarbil (OM; n = 6; black bars) versus placebo (n = 8; grey bars) infusion. Values are presented as mean ± SD; * 0.01 < *p* < 0.05, ** 0.001 < *p* < 0.01, *** *p* < 0.001.

**Table 1 genes-14-00122-t001:** Primers used for real-time quantitative polymerase chain reaction (*RTq-PCR*) in rat left ventricular (LV) myocardial tissue.

Genes		Primer Sequences
Glycerol-3-phosphate dehydrogenase (*GAPDH*)	Sense Antisense	5′-AAGATGGTGAAGGTCGGTGT-3′ 5′-ATGAAGGGGTCGTTGATGG-3′
Hypoxanthine guanine phosphoribosyl transferase (*HPRT*)	Sense Antisense	5′-ACAGGCCAGACTTTGTTGGA-3′ 5′-ATCCACTTTCGCTGATGACAC-3′
AMP-activated protein kinase (*Ampk*)	Sense Antisense	5′-TTCGGGAAAGTGAAGGTGGG-3′ 5′-TCTCTGCGGATTTTCCCGAC-3′
Angiotensin-converting enzyme 1 (*ACE1*)	Sense Antisense	5′-AGTGGGTGCTGCTCTTCCTA-3′ 5′-GGAGGCTGTGATGGTTATGG-3′
Angiotensin-converting enzyme 2 (*ACE2*)	Sense Antisense	5′-GCCTTGGAAAATGTGGTAGG-3′ 5′-TTCAGCCAGACAAACAATGG-3′
Angiotensin II receptor type 1a (*Agtr1a* or *AT1*)	Sense Antisense	5′-ACATTCTGGGCTTCGTGTTC-3′ 5′-CATCATTTCTTGGCGTGTTC-3′
Angiotensin II receptor type 2 (*Agtr2* or *AT2*)	Sense Antisense	5′-TGCTCTGACCTGGATGGGTA-3′ 5′-AGCTGTTTGGTGAATCCCAGG-3′
Arachidonate 15-lipoxygenase (*Alox15*)	Sense Antisense	5′-GCACTCTTCCGTCCATCTTG-3′ 5′-GCTTCTCCATTGTTGCTTCCT-3′
ATPase sarcoplasmic/endoplasmic reticulum Ca^2+^ transporting 2 (*Atp2a2* or *Serca2*)	Sense Antisense	5′-GCAGGTCAAGAAGCTCAAGG-3′ 5′-TCTCTGCGGATTTTCCCGAC-3′
Bcl2 associated X apoptosis regulator (*Bax*)	Sense Antisense	5′-CGTGGTTGCCCTCTTCTACT-3′ 5′-TCACGGAGGAAGTCCAGTGT-3′
B-cell lymphoma 2 (*Bcl2*)	Sense Antisense	5′-TTTCTCCTGGCTGTCTCTGAA-3′ 5′-CATATTTGTTTGGGGCAGGT-3′
Bradykinin receptor B1 (*Bdkrb1*)	Sense Antisense	5′-AAGCTACGTGCCTGCTCATC-3′ 5′-CGGGGACGACTTTAACAGAG-3′
Bradykinin receptor B2 (*Bdkrb2*)	Sense Antisense	5′-GCTGTCGTGGAAGTGGCTAT-3′ 5′-AAGGTCCCGTTATGAGCAGA-3′
Ca^2+^/calmodulin-dependent protein kinase II delta (*Camk2d*)	Sense Antisense	5′-ATCCACAACCCTGATGGAAA-3′ 5′-GCTTTCGTGTTTCACGTCT-3′
Ca^2+^ voltage-gated channel subunit alpha1 C (*Cacna1c*)	Sense Antisense	5′-CCTATTTCCGTGACCTGTGG-3′ 5′-GGAGGGACTTGATGGTGTTG-3′
Carnitine palmitoyltransferase 1 (*Cpt1*)	Sense Antisense	5′-AAGAACACGAGCCAACAAGC-3′ 5′ACCATACCCAGTGCCATCAC-3′
CD36 fatty acid transporter (*Cd36*)	Sense Antisense	5′-TTTCTGCTTTCTCATCGCCG-3′ 5′-GGATGTGGAACCCATAACTGG-3′
Glutathione peroxidase (*Gpx*)	Sense Antisense	5′-CCGACCCCAAGTACATCATT-3′ 5′-AACACCGTCTGGACCTACCA-3′
Glutathione-disulfide reductase (*Gsr*)	Sense Antisense	5′-GCCGCCTGAACAACATCTAC-3′ 5′-CTTTTTCCCGTTGACTTCCA-3′
Kallikrein-related peptidase 8 (*Klk8*)	Sense Antisense	5′-CGGAGACAGATGGGTCCTAA-3′ 5′-ATCTCTTGCTCGGGCTCAT-3′
Kallikrein-related peptidase 10 (*Klk10*)	Sense Antisense	5′-GCAGGTCTCCCTCTTCCATA-3′ 5′-CAGTGGCTTATTTCTCCAGCA-3′
Kallikrein 1-related peptidase C2 (*Klk1c2*)	Sense Antisense	5′- CAGGAGAGATGGAAGGAGGA-3′ 5′-CGGTGTTTTGGGTTTAGCAC-3′
Kallikrein 1-related peptidase C12 (*Klk1c12*)	Sense Antisense	5′-CATCAAAGCCCACACACAGAT-3′ 5′-AAGCACACCATCACAGAGGAG-3′
Nitric oxide synthase 2 (*NOS2* or *iNOS*)	Sense Antisense	5′-GTTTCCCCCAGATCCTCACT-3′ 5′-CTCTCCATTGCCCCAGTTT-3′
Nitric oxide synthase 3 (*NOS3* or *eNOS*)	Sense Antisense	5′-GGTATTTGATGCTCGGGACT-3′ 5′-TGATGGCTGAACGAAGATTG-3′
Oxidized low density lipoprotein receptor 1 (*Olr1* or *Lox1*)	Sense Antisense	5′-CATTCACCTCCCCATTTT-3′ 5′-GTAAAGAAACGCCCCTGGT-3′
Peroxisome proliferator-activated receptor α (*Ppar α*)	Sense Antisense	5′-TTAGAGGCGAGCCAAGACTG-3′ 5′-CAGAGCACCAATCTGTGATGA-3′
Peroxisome proliferator-activated receptor γ (*Ppar γ*)	Sense Antisense	5′– GCGCTAAATTCATCTTAACTC-3′ 5′-CTGTGTCAACCATGGTAATTT-3′
Pyruvate dehydrogenase kinase 4 (*Pdk4*)	Sense Antisense	5′-GAGCCTGATGGATTTAGTGGA-3′ 5′-CGAACTTTGACCAGCGTGT-3′
Ryanodine receptor 2 (*Ryr2*)	Sense Antisense	5′-GGAACTGACGGAGGAAAGTG-3′ 5′-GAGACCAGCATTTGGGTTGT-3′
Solute carrier family 2 member 1 (*Slc2a1* or *Glut1*)	Sense Antisense	5′-TCTTCGAGAAGGCAGGTGTG-3′ 5′-TCCACGACGAACAGCGAC-3′
Solute carrier family 2 member 4 (*Slc2a4* or *Glut4*)	Sense Antisense	5′-AGGCCGGGACACTATACCC-3′ 5′-TCCCCATCTTCAGAGCCGAT -5′
Solute carrier family 8 member A1 (*Slc8a1*)	Sense Antisense	5′-GAGATTGGAGAACCCCGTCT-3′ 5′-AGTGGCTGCTTGTCATCGTA-3′
Superoxide dismutase 1 (*Sod1*)	Sense Antisense	5′-GGTCCACGAGAAACAAGATGA-3′ 5′-CAATCACACCACAAGCCAAG-3′
Superoxide dismutase 2 (*Sod2*)	Sense Antisense	5′-AAGGAGCAAGGTCGCTTACA-3′ 5′-ACACATCAATCCCCAGCAGT-3′

**Table 2 genes-14-00122-t002:** Echocardiographic measurements in rats (n = 6) at baseline and 30 min after OM infusion.

Parameters	Before OM Infusion	After OM Infusion
FS (%)	38.8 ± 3.6	44.1 ± 4.4 *
EF (%)	76.0 ± 4.4	80.6 ± 5.2
LVESD (mm)	5.5 ± 0.5	4.6 ± 0.9
LVEDD (mm)	9.0 ± 0.4	8.3 ± 0.9
HR (beats/min)	283 ± 57	303 ± 68
SBP (mmHg)	128 ± 9	128 ± 21
DBP (mmHg)	90 ± 10	88 ± 20
LVET (ms)	79 ± 6	89 ± 8 *
PEP (ms)	21 ± 5	14 ± 7 *
PEP/LVET	0.26 ± 0.06	0.15 ± 0.07 *
CO (mL/min)	90 ± 21	106 ± 22
SV (mL)	0.32 ± 0.06	0.35 ± 0.04
LA (mm)	5.8 ± 0.8	5.7 ± 0.5

Values presented are means ± SD; * *p* < 0.05. FS, fractional shortening; EF, ejection fraction; LVESD, left ventricle end-systolic diameter; LVEDD, left ventricle end-diastolic diameter; HR, heart rate; PEP, pre-ejection period; LVET, left ventricular ejection time; SV, stroke volume; CO, cardiac output; SBP, systolic blood pressure; DBP, diastolic blood pressure; LA, left atrial diameter.

## Data Availability

All experiments were conducted in the Physiology and Pharmacology Laboratory of the Université Libre de Bruxelles (Brussels, Belgium). All data are available and accessible upon request.

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
