# Peer review of "Altered Left Ventricular Rat Gene Expression Induced by the Myosin Activator Omecamtiv Mecarbil"

_genes, 2023, doi:10.3390/genes14010122_

Round 1
Reviewer 1 Report
In this manuscript (MS), the authors o explore the impact of omecamtiv mecarbil (OM) on gene expression profile in adult male rats. This study demonstrated that A single intravenous infusion of OM, in adult helthy rats, resulted in significant changes in LV expression of genes regulating apoptosis, oxidative stress, metabolism, and cardiac contractility.
The overall level of the manuscript is very good which is investigated an interesting topic. In my opinion, the manuscript is well written. However, a major concern about this study is that the authors should have used an animal model of with left ventricular pathology of problem to compare and study the effect of the drug. The authors should clarify this issue.
In addition, there is a list of some comments and questions on points which could be more elaborated or discussed:
1. The manuscript requires proofreading and revision to improve the quality of English (for example, in line 34, helthy should be corrected). Likewise, abbreviations should be revised.
2. materials and Methods should be revised and modified; Transthoracic 2D, M-mode, and Doppler echocardiography should be in a separate section.
3. Statistical analysis should be revised by an expert in the field.
4. Figures should be re constructed in a clearer way.
5. The authors should have, at least, investigated the histology of the LV or fibrosis area in rats to show the effect of OM.
6. The authors should provide the following information:
Author Contributions:
Funding:
Institutional Review Board Statement:
Informed Consent Statement:
Data Availability Statement:
Conflicts of Interest:
Author Response
Reply to the Reviewer#1
The overall level of the manuscript is very good which is investigated an interesting topic. In my opinion, the manuscript is well written.
Re: We thank the reviewer for these positive comments.
However, a major concern about this study is that the authors should have used an animal model of with left ventricular pathology of problem to compare and study the effect of the drug. The authors should clarify this issue.
Re: We agree that it would be very interesting to evaluate the effects of OM in an experimental model of left ventricular pathology, which we plan to test in future experiments. The aim of the present study was to evaluate first the effects of OM on a healthy heart, before an evaluation in a pathological condition. Indeed, it allowed us to better understand the mechanisms underlying the effects of OM, which were not known. This has been added in the Discussion section (see page 16).
In addition, there is a list of some comments and questions on points which could be more elaborated or discussed:
- The manuscript requires proofreading and revision to improve the quality of English (for example, in line 34, helthy should be corrected). Likewise, abbreviations should be revised.
R1: We thank the reviewer for these comments. A linguistic and grammatical proofreading, as well as a revision of abbreviations and acronyms, were carried out and corrections were made accordingly throughout the manuscript. We hope that this is now better and clearer throughout the manuscript.
- Materials and Methods should be revised and modified; Transthoracic 2D, M-mode, and Doppler echocardiography should be in a separate section.
R2: Agreed. Methods for echocardiography and cardiac measurements have been added and better detailed in the Material & Method section (see pages 6-7).
- Statistical analysis should be revised by an expert in the field.
R3: We apologize for the insufficient clarity and description of the statistical method and analysis. This has been accordingly modified in the Method-Statistical analysis section (see page 8). This was revised and adapted with the help of Professor Christian Melot, Professor of Biostatistics at the Faculty of Medicine of the Université Libre de Bruxelles (Brussels, Belgium). We therefore would like to add Pr Melot as co-author of the present work.
- Figures should be re constructed in a clearer way.
R4: We thank the reviewer for this judicious comment. The figures have been reworked in a clearer manner and in respect with the standard of the Genes journal. We hope that the figures are now clearer.
- The authors should have, at least, investigated the histology of the LV or fibrosis area in rats to show the effect of OM.
R5: We thank the reviewer for this pertinent suggestion. Unfortunately for technical reasons beyond our control, we were unable to collect left ventricular samples for histological examination. This will definitely be done in future experiments with OM test.
- The authors should provide the following information:
Author Contributions: All experiments were conducted in the Physiology and Pharmacology Laboratory of the University of Brussels. B.E. conceived the study, participated in its design, carried out the experiments, performed the statistical analysis and drafted the manuscript. K.M.E. conceived and coordinated the study, participated in its design, carried out the echocardiography and read and analyzed them. G.H. participated in the coordination of the study and carried out the experiments. P.J. and C.S. contributed to technical assistance, performed the statistical analysis and reviewed the manuscript. C.M. performed the statistical analysis. P.V.B., L.D. and F.V. conceived the study and participated in the drafting of the manuscript.
Funding: This study was supported and funded by the Fonds pour la Chirurgie Cardiaque, Brussels, Belgium grant number 489639.
Institutional Review Board Statement: The experimental protocol was approved by the Institutional Animal Care and Use Committee of the Université Libre de Bruxelles (Brussels, Belgium). Studies were conducted in accordance with the Guide for the Care and Use of Laboratory Animals published by the National Institutes of Health (NIH Publication No. 85-23; revised 1996).
Informed Consent Statement: All authors read and approved the final version of the manuscript.
Data Availability Statement: All experiments were conducted in the Physiology and Pharmacology Laboratory of the Université Libre de Bruxelles (Brussels, Belgium). All data are available and accessible upon request.
Conflicts of Interest: None declared.
Dear Reviewer 1,
As suggested, we provided additional information in the Methods sections, and we clarified the Figures. The manuscript has been thoroughly proofread.
We respectfully hope that this revised version of our manuscript might be found acceptable.
We thank the reviewer for the valid comments and criticisms, which were very helpful to improve our work.
Yours sincerely,
Reviewer 2 Report
Despite Omecamtiv mecarbil (OM) is only over 5 years in therapy of acute heart condition there is no fully described mechanism of functional effect of this medication to cardiac tissue. Authors made a good effort to observe expression of wide range of genes. Authors accept limitations of this study and it is good approach to continue to explore other models and designs of experiments in the future studies to investigate accumulative effect of genes in respond to OM.
Author Response
Reply to Reviewer #2
Despite Omecamtiv mecarbil (OM) is only over 5 years in therapy of acute heart condition there is no fully described mechanism of functional effect of this medication to cardiac tissue. Authors made a good effort to observe expression of wide range of genes. Authors accept limitations of this study and it is good approach to continue to explore other models and designs of experiments in the future studies to investigate accumulative effect of genes in respond to OM.
R: We respectfully thank the reviewer for these positive comments. We fully agree that the present study would be completed by future studies evaluating the effects of OM in pathological conditions, more specifically those impacting the left ventricle. This is what we plan to test in future studies. This is now added in the Discussion section (see page 16). However, we first started evaluating the effects of OM infusion on a healthy heart. To our knowledge, this is the first study evaluating that point at pathobiological level.
Reviewer 3 Report
Dear Editor,
I have had the pleasure of reading this manuscript well-written and explained step by step. In my opinion, this paper shows high standards of quality. I would suggest a minor revision of the grammar
Regards
Author Response
Reply to Reviewer #3
I have had the pleasure of reading this manuscript well-written and explained step by step. In my opinion, this paper shows high standards of quality.
R: We thank the reviewer for these positive comments.
I would suggest a minor revision of the grammar
R: A linguistic and grammatical proofreading was performed and corrections were made accordingly throughout the manuscript. We hope that this is now better.
Round 2
Reviewer 1 Report
The authors have responded to my comments. Therefore, the ms can be accepted now.